# Self-compliant ionic nanomesh for gas-permeable and stress-free on-skin electronics

Qinqing Du, Lingyan Liu, Shengtong Sun ⊠ & Peiyi Wu ⊠

Self-adaptive compliance and gas permeability are crucial properties for on-skin electronics, enabling the reliable extraction of high-fidelity electrophysiological signals for applications in personal healthcare and robotic control. However, integrating these two properties into a single device, particularly under dynamic skin deformations, remains a significant challenge. Here, we present an ultrathin liquid crystal elastomer-based sheath-core ionic nanomesh that synergizes soft elasticity with ionic conductivity to create self-compliant and breathable bioelectronic interfaces. The nanomesh features a hydrophilic sheath and a porous architecture, which ensure high moisture and air permeability for superior wearable comfort. Moreover, the unique liquid-like deformation of liquid crystal directors facilitates nearly stress-free skin-device junctions, promoting fatigue-resistant adhesion against various interfacial failures. Our fabricated electrodes successfully acquire muscle-specific electromyography signals with minimized motion artifacts - a feat challenging for conventional epidermal electrodes. This design resolves the trade-off between self-compliance and permeability, establishing a new paradigm for long-term, reliable wearable electronics.

The emergence of on-skin electronics for continuous electrophysiological monitoring has unlocked profound capabilities in personal healthcare and robotic control applications[1–5]. Long-term wearability and accurate signal acquisition require devices to be both compliant and gas-permeable to minimize skin inflammation and interfacial stress. Significant progress has been made toward this goal through various approaches, including the development of nanomeshes[6–10], ultrathin elastomer/hydrogel films[11–15], metamaterials[16], and serpentine/fibrous electrodes[3,17,18]. In particular, thinning porous or hollow devices can simultaneously lower their effective modulus and gas-permeable resistance, significantly enhancing both skin conformability and breathability. However, despite the effectiveness of these methods under resting conditions, acquiring reliable on-skin signals under dynamic conditions remains a considerable challenge.

Human skin is constantly moving and deforming in various ways and at different frequencies. This continuous movement generates interfacial stress between electronic devices and skin, often leading to altered performance, poor contact, and motion artifacts. This problem becomes even more pronounced when devices are placed on highly mobile and intricate skin surfaces, such as human hands. In these situations, muscle-specific electromyography (EMG) can monitor precise hand actions, a crucial capability for applications in rehabilitation, augmented reality, and robotic control[19,20]. However, a major challenge remains: integrating self-adaptive compliance and gas permeability into a single on-skin device that performs reliably under these dynamic conditions.

Signal filtering is a common method for removing motion artifacts, but it has drawbacks such as high power consumption, data delays, and the difficulty of building accurate classification models[21]. While using ultrasoft materials or improving interfacial adhesion can somewhat alleviate these problems[22–25], mechanical elastic energy still accumulates during long-term deformations, causing eventual stress-induced delamination. A more effective strategy was recently proposed that involves imparting high viscosity to the substrate material, for instance, by adjusting its rheology to a critical gel point state[26–30].

State Key Laboratory of Advanced Fiber Materials, College of Chemistry and Chemical Engineering & Center for Advanced Low-dimension Materials, Donghua University, Shanghai, China. ⊠e-mail: shengtongsun@dhu.edu.cn; wupeiyi@dhu.edu.cn

This approach can dampen concentrated stress, enabling self-adaptive compliance for long-term, high-fidelity electrophysiological signal detection. Unfortunately, materials with such high viscosity often self-fuse and typically form non-porous gels, precluding gas permeability.

Liquid crystal elastomers (LCEs) are a class of smart materials that uniquely combine the orientational order of liquid crystals with the deformability of elastomer networks[31,32]. This coupling gives rise to a distinctive liquid-like deformation—termed "soft elasticity"—even at high strains (up to 50%), attributed to the nearly zero-stress reorientation of the LC directors[33–35]. The soft elasticity of LCEs has been extensively explored for advanced pressure-sensitive

adhesives, enabling self-compliant adhesion through stress-adaptive conformal contact[36–39]. Crucially, chemical crosslinking in LCEs can prevent self-fusion without sacrificing their ultrahigh viscosity. This is because LCE viscosity is governed by the free rotation of mesogen groups rather than by the self-fusible free and dangling chains[40,41]. We anticipate that creating an ultrathin porous LCE film will over-come the trade-off between self-compliance and gas permeability under dynamic conditions.

Here, we introduce a coaxial electrospinning strategy to fabricate an ultrathin LCE-based ionic nanomesh that effectively combines self-compliance and gas permeability (Fig. 1a). Each nanofiber within this

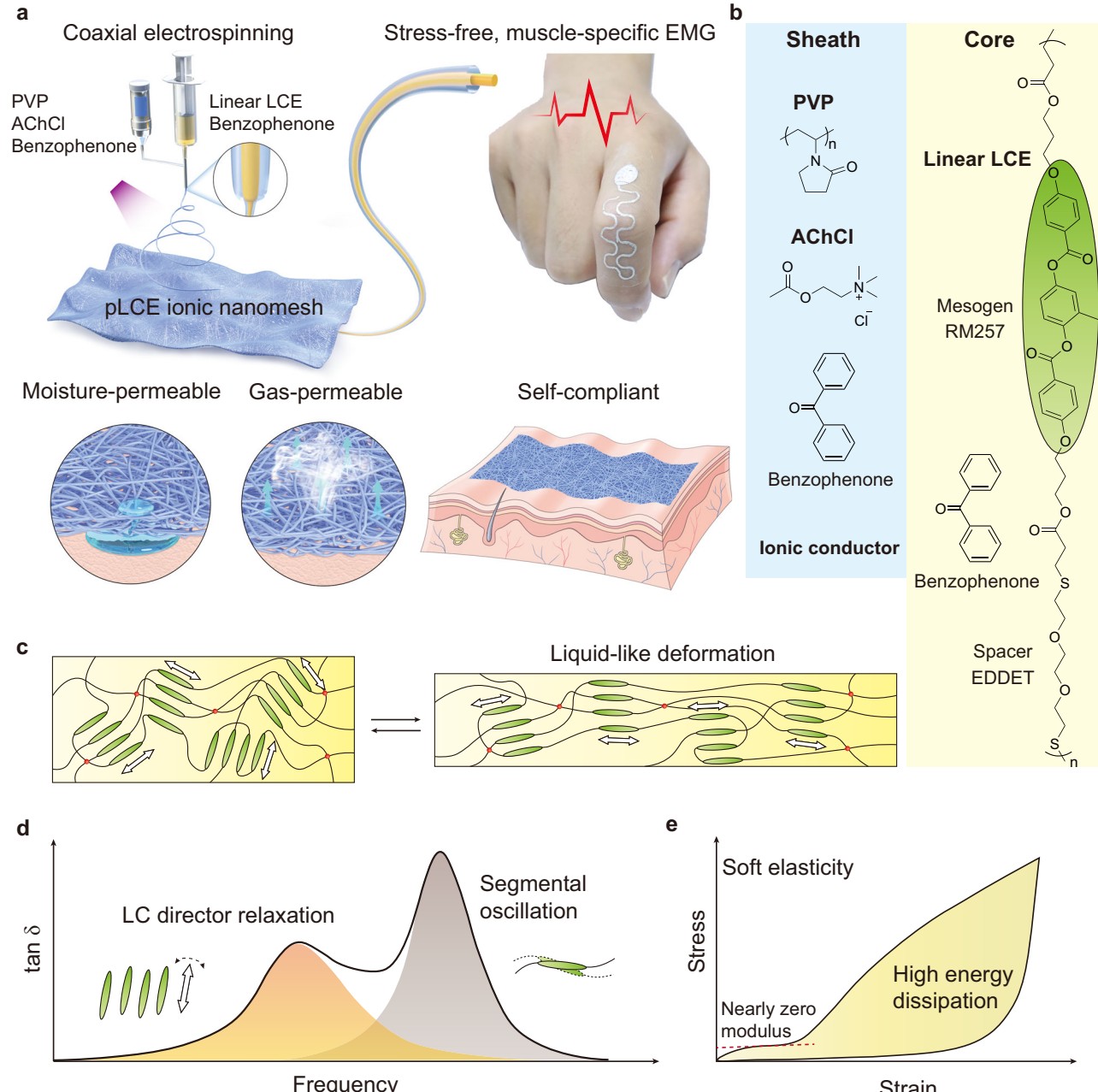

**Fig. 1 | Working mechanism of self-compliant and permeable liquid crystal elastomer (pLCE) ionic nanomesh. a** Coaxial electrospinning setup, properties and application of the ultrathin pLCE ionic nanomesh. UV-sensitive benzophenone was incorporated into both the core and sheath spinning dopes to induce chemical crosslinking during electrospinning. **b** Main chemical components within the core and sheath layers. **c** Schematic of the liquid-like deformation of LCE chains via the free rotation of LC directors. **d** A typical loss factor (tan δ) curve for LCE, exhibiting

two distinct peaks corresponding to LC director relaxation and segmental oscillation (or glass transition). **e** Soft elasticity of LCE, featuring nearly zero-modulus deformation and high energy dissipation when stretched. This effect minimizes the interfacial stress at the skin-device junction, providing pLCE ionic nanomesh with excellent self-compliance for muscle-specific electromyography (EMG) monitoring, even during dynamic skin deformations.

permeable LCE (pLCE) ionic nanomesh features a sheath-core structure: a self-compliant LCE core is encapsulated by a biocompatible, ultrasoft ionic sheath composed of polyvinylpyrrolidone/acetylcholine chloride (PVP/AChCl) (Fig. 1b). The LCE's soft elasticity and liquid-like deformation generate an additional loss factor (tan δ) peak, resulting from LC director relaxation (Fig. 1c, d). This unique property enables nearly zero-modulus deformation with high energy dissipation, effectively reducing the interfacial stress between the ionic nanomesh and human skin during dynamic motions (Fig. 1e). When integrated with stretchable circuits and applied to human hands, the fabricated ionic nanomesh electrodes are capable of capturing high-quality, muscle-specific EMG signals, allowing for the precise classification of even very subtle thumb actions.

## Results

### Fabrication and characterization of pLCE ionic nanomesh

We prepared pLCE ionic nanomesh using an ultraviolet (UV)-assisted coaxial electrospinning process. For the core flow, linear long LCE chains were synthesized via a Michael addition reaction between an acrylate-terminated reactive mesogen (RM257) and a thiol-terminated chain spacer (2,2′-(ethylenedioxy)diethanethiol, EDDET), yielding a number-average molecular weight of $3.4 \times 10^4$ g mol$^{-1}$ (Supplementary Figs. 1 and 2). To improve the spinnability of the core flow, a small proportion (5.6 wt%) of thermoplastic polyurethane (TPU) was added to the LCE solution (solvent: dimethylformamide/tetrahydrofuran, DMF/THF). For the sheath flow, PVP and AChCl were dissolved in a mixture of DMF and ethanol. UV-sensitive benzophenone was incorporated into both the core and sheath flows to induce rapid chemical crosslinking during electrospinning, thereby curing the resulting ionic nanomesh. Upon UV irradiation, benzophenone molecules are activated and abstract hydrogen atoms from the LCE and PVP chains to generate radicals for both interchain and interfacial coupling (Supplementary Fig. 3)[42]. The resulting ionic nanomesh is remarkably thin (typically only 8 μm), enabling it to conform to microcurvatures (Supplementary Fig. 4).

We highlight that the sheath-core design is crucial for creating this self-compliant ionic nanomesh. While directly electrospinning an LCE ionogel might seem more straightforward, the presence of a highly polar ionic liquid significantly degraded the quality of the electrospun nanofibers (Supplementary Fig. 5). Furthermore, the introduced ionic liquid often plasticized the LCE network, causing the characteristic soft elasticity peak to diminish or even disappear entirely (Supplementary Fig. 6). In contrast, the sheath-core design effectively decouples the functions of soft elasticity and ionic conductivity into two distinct layers, enabling the fabrication of a high-quality, self-compliant ionic nanomesh. Notably, the sheath material in this structure must be ultrasoft to avoid interfering with the soft elasticity of the core material.

The coaxial electrospinning process allowed for precise control over the architecture, porosity, and ionic conductivity of the resulting ionic nanomesh. We maintained a core flow rate of 0.3 mL h$^{-1}$, while varying the sheath flow rate from 0 to 1.2 mL h$^{-1}$. With only the core flow, the resulting neat LCE nanomesh exhibited a typical fiber diameter of ~300 nm and an areal porosity of 43%, as determined by scanning electron microscopy (SEM) (Fig. 2a, b). Increasing the sheath flow rate led to the growth of the PVP/AChCl ionic sheath, and the interfibrillar gaps were progressively filled. Increasing the sheath flow rate from 0.2 to 1.2 mL h$^{-1}$ produced an ionic sheath with an average thickness ranging from 200 nm to 1.2 μm, and a corresponding reduction in areal porosity from 37% to 16% (Fig. 2b, c). Further increasing the sheath flow rate to 1.4 mL h$^{-1}$ led to an excessively thick sheath layer, resulting in a nearly complete loss of nanomesh porosity (Supplementary Fig. 7). Similarly, when the sheath flow rate was maintained at 0.4 mL h$^{-1}$, increasing the core flow rate from 0 to 0.9 mL h$^{-1}$ led to thicker LCE cores, gradually reducing the areal porosity from 49% to 23% (Supplementary Fig. 8). For a representative

ionic nanomesh produced at core and sheath flow rates of 0.3 and 0.4 mL h$^{-1}$, respectively, no fiber self-fusion was observed over time, indicating a stable, chemically crosslinked porous architecture that is crucial for long-term gas permeability. Notably, the applied voltage and collector rotation speed also strongly influenced the spinning quality. A voltage or collector rotation speed that was too high led to excessively fast nanofiber deposition, leaving insufficient time for uniform UV crosslinking, and the resulting fibers tended to self-fuse (Supplementary Figs. 9 and 10).

The successful integration of the core and sheath components was confirmed by Fourier transform infrared (FTIR) spectral comparison (Supplementary Fig. 11). Furthermore, the sheath-core structure was supported by elemental mapping of S atoms (exclusive to LCE in the core) and Cl atoms (exclusive to AChCl in the sheath) using transmission electron microscopy (TEM) (Fig. 2d). Complementary evidence was provided by confocal laser scanning microscopy (CLSM) imaging of the stained sheath (Fig. 2e). Finally, polarized optical microscopy (POM) revealed the liquid crystalline state of the LCE core under ambient conditions, as shown by the uniform bright field of the ionic nanomesh (Fig. 2f). The ambient liquid crystalline state was also confirmed by the heat-induced rapid contraction of a stretched pLCE ionic nanomesh (Supplementary Fig. 12).

Owing to the hygroscopic nature of PVP and AChCl, varying the sheath flow rate profoundly influences the water adsorption properties of the pLCE ionic nanomesh, which in turn is vital for achieving moisture permeation and ionic conductivity. In contrast to neat LCE nanomesh, which was hydrophobic and showed no water adsorption, increasing the sheath flow rate from 0.2 to 1.2 mL h$^{-1}$ led to a dramatic increase in water content from 3 wt% to 11.2 wt% (Fig. 2g; original data in Supplementary Fig. 13). To further investigate the mobility of polymers and water molecules within the ionic nanomesh, we employed low-field $^1$H nuclear magnetic resonance (NMR) spectroscopy (Fig. 2h). Our analysis showed that the $T_2$ (spin-spin relaxation time) peak corresponding to polymer chains remained consistent despite variations in sheath flow rates. However, a distinct $T_2$ peak corresponding to adsorbed water emerged at approximately 10 ms. This suggests that a thicker ionic sheath facilitates the adsorption of more water molecules, which exhibit higher mobility. As a direct consequence, we observed a significant enhancement in ionic conductivity, rising from an ultralow value of $3.9 \times 10^{-8}$ S m$^{-1}$ for the neat pLCE nanomesh to $1.3 \times 10^{-4}$ S m$^{-1}$ for the pLCE ionic nanomesh fabricated at a sheath flow rate of 1.2 mL h$^{-1}$ (Fig. 2i).

### Moisture and gas permeability

Permeability to both moisture and gas directly impacts the long-term comfort and practicability of wearable electronics. We measured the water vapor transmission rates (WVTRs) of pLCE ionic nanomeshes using a wet-cup method under ambient conditions (25 °C, RH 50%). For comparison, we used a cast PVP/AChCl film and a common polydimethylsiloxane (PDMS) film, both approximately 100 μm thick, as control samples. Our pLCE ionic nanomeshes consistently achieved a WVTR exceeding 1200 g m$^{-2}$ day$^{-1}$ across all sheath flow rates (Fig. 3a). This value far surpasses the typical daily physiological sweat evaporation rate (~600 g m$^{-2}$ day$^{-1}$) and is comparable to the reported best-performing breathable on-skin materials (see comparison in Supplementary Table 1). This suggests that the introduction of the PVP/AChCl ionic sheath did not significantly impair the nanomesh's moisture permeability, even as porosity decreased with increasing sheath flow rates. In comparison, the nonporous PVP/AChCl film showed a reduced WVTR of 923 g m$^{-2}$ day$^{-1}$, while the hydrophobic PDMS film showed a much lower WVTR of 363 g m$^{-2}$ day$^{-1}$.

We further evaluated the air permeability of pLCE ionic nanomeshes using the differential pressure method (Fig. 3b). All nanomeshes showed a quasi-linear increase in air permeability with increasing pressure drop, indicating stable porous structures resistant

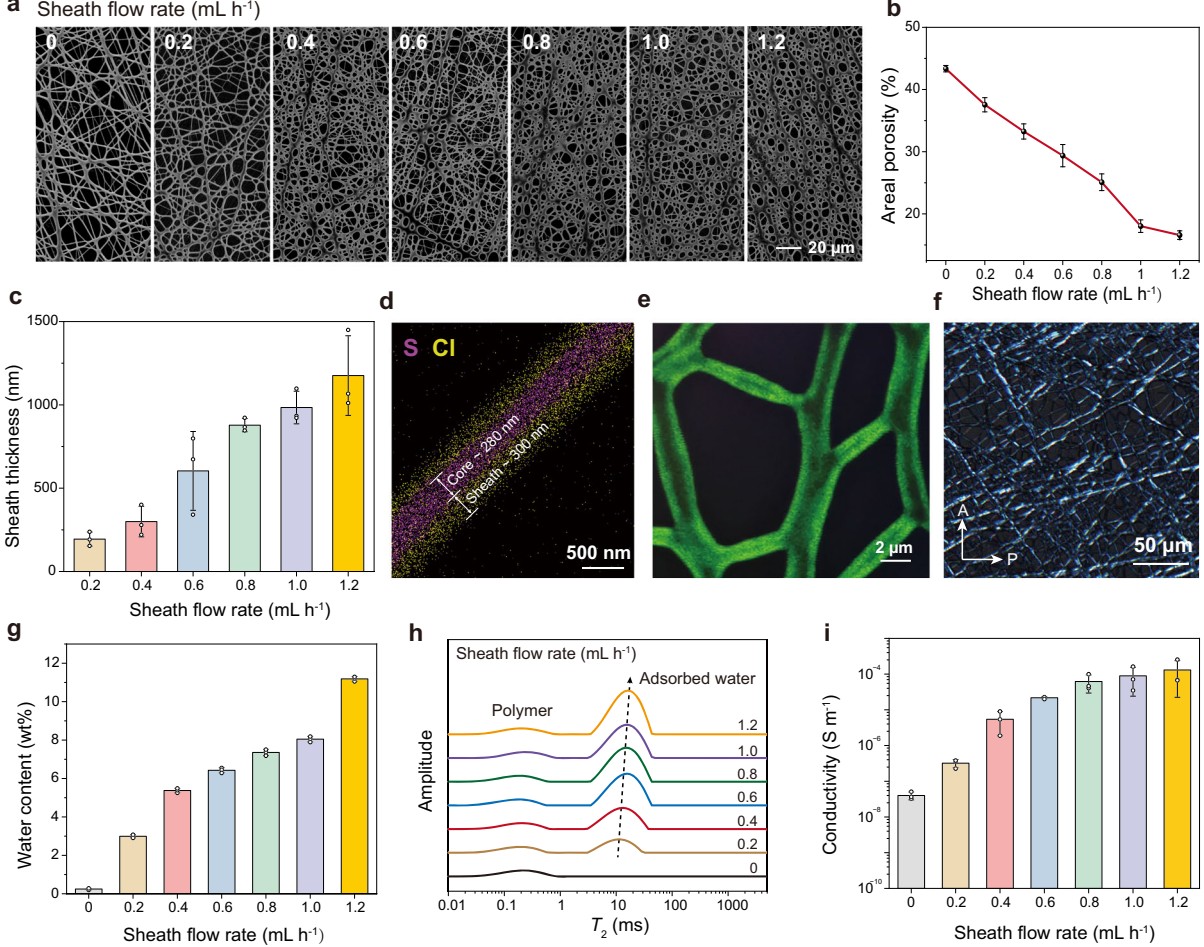

**Fig. 2 | Fabrication and characterization of pLCE ionic nanomesh. a** SEM images of ionic nanomeshes fabricated at varying sheath flow rates (0-1.2 mL h$^{-1}$; core flow rate = 0.3 mL h$^{-1}$). **b, c** Calculated areal porosity and sheath layer thickness, respectively. **d** A typical TEM elemental mapping image illustrating the sheath-core structure of the nanofibers. **e** CLSM image of the ionic nanomesh, with the sheath fluorescently labeled using Coumarin 6. **f** POM image of the ionic nanomesh, revealing the liquid crystalline state of the LCE core. **g** Dependence of water content on sheath flow rate. **h** Solid-state low-field $^1$H NMR spectra showing water and polymer mobility at 25 °C and RH 50%. **i** Dependence of ionic conductivity on sheath flow rate. Nanomeshes for (d-f) were all fabricated at a sheath flow rate of 0.4 mL h$^{-1}$. Data are presented as the mean values ± SD, $n$ = 3 independent samples. Source data are provided as a Source Data file.

to deformation. As expected, ionic nanomeshes with higher porosity (prepared with lower sheath flow rates) exhibited higher air permeability. In contrast, the nonporous cast PVP/AChCl film showed almost no air permeability. A typical pLCE ionic nanomesh fabricated at a sheath flow rate of 0.4 mL h$^{-1}$ had an air permeability of 471 L m$^{-2}$ s$^{-1}$ at a 100 Pa pressure drop, comparable to common woven fabrics[43]. All these results highlight the critical role of the ionic nanomesh's porous architecture in enabling rapid moisture and air permeation.

To visually demonstrate the pLCE ionic nanomesh's permeability, we generated a "water cloud" by combining dry ice and hot water in a small bottle. This cloud readily permeated through the nanomesh covering the bottle (Fig. 3c and Supplementary Movie 1). Furthermore, when inflated underwater, the covered pLCE ionic nanomesh could "breathe" much like a frog's vocal sac (Supplementary Movie 2). The deformed nanomesh also exhibited significant water retention without leakage (Fig. 3d). Yet, upon contact with a water surface, rapid water permeation occurred, and the nanomesh instantaneously recovered its original undeformed state. This swift water penetration was driven by its hydrophilic surface, as evidenced by low contact angles for both water and sweat (39–45°) (Fig. 3e) and high-speed camera observation of the water transport process (Supplementary Movie 3). In contrast, hydrophobic liquids like ethyl acetate and corn oil could not readily permeate the ionic nanomesh (Supplementary Fig. 14).

We used 2D low-field $^1$H NMR to further elucidate the moisture permeation mechanism. In the 2D $T_1$-$T_2$ correlation map ($T_1$: spin-lattice relaxation time; $T_2$: spin-spin relaxation time), a lower $T_1/T_2$ ratio generally signifies higher molecular mobility, with the diagonal line ($T_1/T_2 = 1$) representing the fully mobile state limit[44]. The 2D low-field $^1$H NMR spectrum of the pLCE ionic nanomesh, after equilibrating at RH 90%, showed four distinct peaks with decreasing $T_1/T_2$ ratios. These peaks were attributed to the polymer, confined water, intermediate water, and mobile water, respectively (Fig. 3f). This suggests that moisture permeates through the ionic nanomesh via a three-level transport mechanism: water moving through interfibrillar pores (mobile water), sorption at the water-polymer interface (intermediate water), and liquid transport within the polymer network (confined water)[45]. As humidity increased, the water peak broadened and shifted to a higher $T_2$, indicating a growing proportion of mobile water transport (Supplementary Fig. 15).

## Adhesive, mechanical, and self-compliant properties

The sheath-core structure of the pLCE ionic nanomesh and the soft elasticity of the LCE endow it with excellent adhesion, high energy dissipation, and dynamic self-compliance. Strong interfacial adhesion is crucial for maintaining close contact between on-skin electronic devices and human skin. We first evaluated the interfacial toughness of

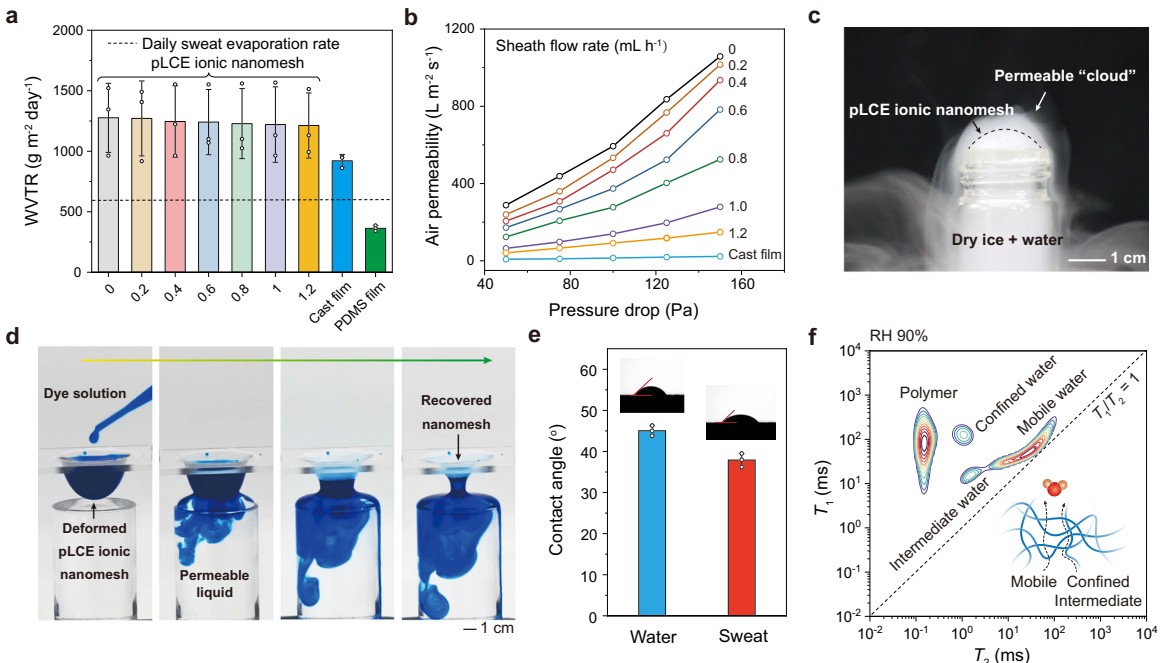

**Fig. 3 | Moisture and gas permeability of pLCE ionic nanomesh. a** Water vapor transmission rates (WVTRs) of pLCE ionic nanomeshes fabricated at varying sheath flow rates, compared to a PVP/AChCl cast film and a PDMS film. **b** Air permeability under incremental pressure drops. **c** Demonstration of the water cloud permeability of pLCE ionic nanomesh. **d** Demonstration of water permeability (liquid stained by methylene blue). **e** Contact angles of water and sweat on pLCE ionic nanomesh. **f** Solid-state 2D low-field $^1$H NMR spectrum of pLCE ionic nanomesh equilibrated at RH 90%. The inset diagram illustrates how confined/intermediate water and mobile water are transported through pLCE ionic nanomesh. Nanomeshes for (c-f) were all fabricated at a sheath flow rate of 0.4 mL h$^{-1}$. Data are presented as the mean values ± SD, $n = 3$ independent samples. Source data are provided as a Source Data file.

pLCE ionic nanomeshes, fabricated at different sheath flow rates, using a 90° peeling test on a glass substrate (Fig. 4a). All nanomeshes, even those without an ionic sheath, exhibited an interfacial toughness exceeding 150 J m$^{-2}$, highlighting the key role of the LCE's soft elasticity in achieving pressure-sensitive adhesion[36,37]. A maximum interfacial toughness of 370 J m$^{-2}$ was achieved at a sheath flow rate of 0.4 mL h$^{-1}$, where the nanomesh has a 300 nm-thick ionic sheath. Both decreasing and increasing the sheath thickness led to reduced interfacial adhesion. Compared to the viscoelastic LCE core, which has a Young's modulus of 326 kPa, the PVP/AChCl sheath is significantly softer, with a Young's modulus of 44 kPa, and exhibits a more elastic response (Supplementary Fig. 16). While the soft, elastic ionic sheath facilitates good interfacial contact with substrates, an overly thick sheath compromises the overall energy dissipation of the nanomesh. Consequently, unless otherwise stated, all discussions of pLCE ionic nanomeshes refer to the one fabricated at a sheath flow rate of 0.4 mL h$^{-1}$. This optimized nanomesh consistently supported a 10 g weight on a glass substrate for over 3 days (Fig. 4b). High interfacial toughness was also evident on diverse substrates, including aluminum (Al), polyethylene terephthalate (PET), ceramic, polymethyl methacrylate (PMMA), skin, and polytetrafluoroethylene (PTFE) (Fig. 4c; porcine skin adhesion shown in Supplementary Fig. 17).

We then measured the tensile stress-strain curve of pLCE ionic nanomesh, which exhibited a distinct soft elasticity region characterized by a low-modulus plateau spanning 50% strain (Fig. 4d). As previously stated, this soft elasticity arises from the nearly stress-free reorientation of LC directors. Further stretching of the ionic nanomesh led to a pronounced strain-stiffening effect, a skin-like mechanical response for self-protection under high strains[46]. This phenomenon is visually corroborated by the almost identical deformation ratio when the nanomesh supported 1 g and 4 g steel balls (Fig. 4e). A maximum elongation was observed at 360% strain (Supplementary Fig. 18). Upon unloading from 200% strain, the nanomesh demonstrated full elastic

recovery with very high energy dissipation, indicated by a hysteresis ratio up to 83%. Moreover, owing to its nanofibrous structure, the pLCE ionic nanomesh is also highly notch-resistant (Supplementary Fig. 19), demonstrating its robustness against mechanical damage.

We then performed time-temperature superposition rheology to investigate the viscoelasticity of the LCE at 25 °C (Fig. 4f). For accurate measurement, an LCE film with a composition identical to that of the LCE core was prepared. As frequency increased, three distinct regions were observed: rubbery, dissipating, and glassy. However, unlike typical viscoelastic polymers that exhibit only one tan δ peak attributed to the glass transition (or segmental oscillation)[47], a second tan δ peak, clearly ascribed to LCE soft elasticity, was discerned. The tan δ peak value from soft elasticity reached as high as 0.8, indicating its high energy-dissipating nature for stress relaxation. Crucially, this peak covers the typical frequency range of human motion (0.1–50 Hz), which is vital for suppressing motion artifacts in on-skin electronic applications. The temperature-sweep rheology curve of the LCE film also confirmed the presence of the soft elasticity region at room temperature (25 °C; Supplementary Fig. 6).

To demonstrate the advantage of pLCE ionic nanomesh for stress-free self-compliance, we prepared another control sample: a permeable TPU (pTPU) ionic nanomesh with the same PVP/AChCl ionic sheath but a pure TPU core. The pLCE ionic nanomesh exhibited ultrarapid stress relaxation, with a relaxation time of merely 51 s (Fig. 5a), a value shorter than that of many biological tissues[48]. In stark contrast, the pTPU ionic nanomesh exhibited a more elastic response, characterized by much slower stress relaxation. The inherently elastic nature of pTPU ionic nanomesh was further confirmed by its tensile loading-unloading and temperature-sweep rheological curves (Supplementary Fig. 20).

The high stress relaxation behavior of pLCE ionic nanomesh enabled stress-free adhesion on almost any deformed substrates. For

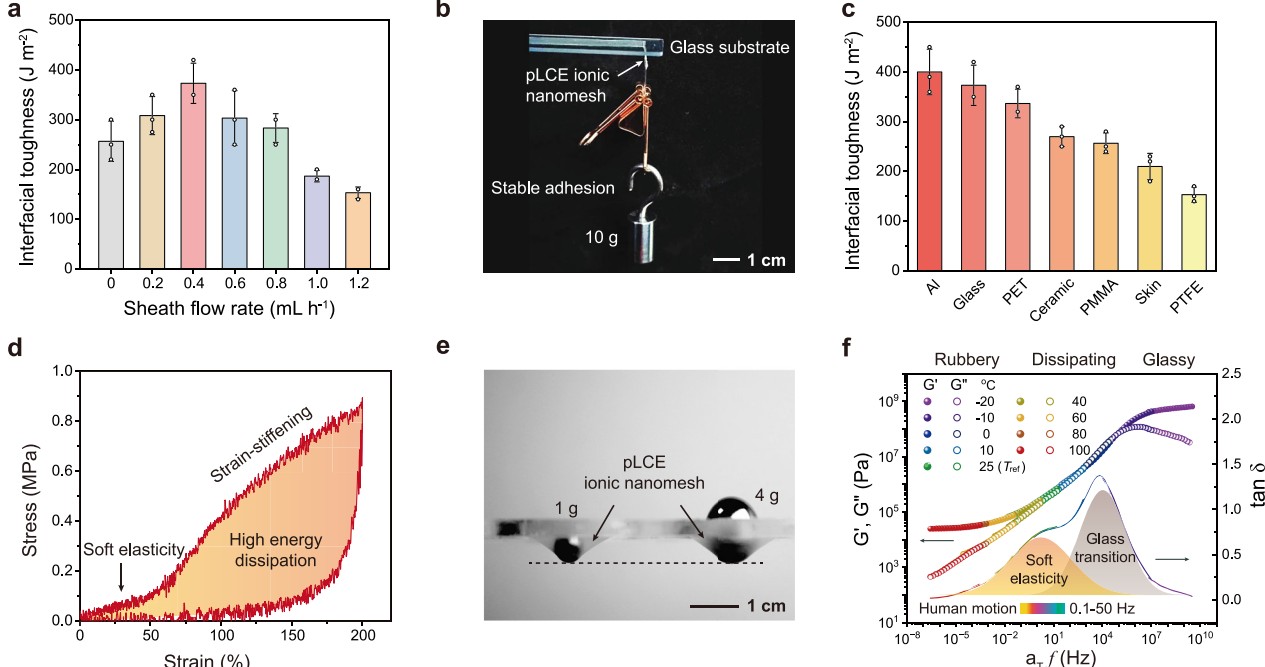

**Fig. 4 | Adhesive and mechanical properties of pLCE ionic nanomesh.**
**a** Interfacial toughness of pLCE ionic nanomeshes fabricated at varying sheath flow rates (substrate: glass). **b** Photograph of a pLCE ionic nanomesh adhered to a glass substrate supporting a 10 g weight for 3 days. **c** Interfacial toughness on various substrates. **d** Tensile loading-unloading curve of pLCE ionic nanomesh. **e** Photograph of the nanomesh supporting 1 g and 4 g weights, visually corroborating strain-stiffening. **f** Time-temperature superposition rheological curves for an LCE film at a reference temperature of 25 °C. Data are presented as the mean values ± SD, $n = 3$ independent samples. Source data are provided as a Source Data file.

instance, we adhered both pLCE and pTPU ionic nanomeshes to a styrene-ethylene-butylene-styrene block copolymer (SEBS) substrate. After several cycles of stretching and releasing the substrate, no interfacial failure occurred with pLCE ionic nanomesh, while delamination readily appeared with the elastic pTPU ionic nanomesh (Fig. 5b and Supplementary Movie 4). This self-compliant adhesion was further observed by attaching pLCE ionic nanomesh to an inflating balloon, where it endured up to 1200% biaxial areal strain (Fig. 5c). After adhering to human hand skin, repeated rubbing with a finger did not induce delamination, further demonstrating its scratch resistance (Supplementary Fig. 21).

We further compared the fatigue-resistant adhesion of pLCE and pTPU ionic nanomeshes across three different interfacial failure modes (Fig. 5d). In small-amplitude 90° peeling tests on a glass substrate, the peeling displacement of pLCE ionic nanomesh remained stable at 1.2 mm under a constant peeling strength of 20 J m⁻² over 1000 cycles. In stark contrast, pTPU ionic nanomesh failed under the same repeated peeling conditions, showing a significant increase in peeling displacement and notable interfacial failure. Fatigue interfacial failure for pLCE ionic nanomesh was observed only when the peeling strength reached 80 J m⁻² (Supplementary Fig. 22). Cycled lap-shear tests were performed with pLCE and pTPU ionic nanomeshes sandwiched between two porcine skin substrates. At a constant shear strain of ±5%, pLCE ionic nanomesh maintained a high shear stress for 1000 cycles, whereas pTPU ionic nanomesh exhibited a gradually reduced shear stress. Likewise, a stable probe-tack adhesive force was observed for pLCE ionic nanomesh adhered to porcine skin over 500 cycles, while pTPU ionic nanomesh showed a progressive decline in adhesive force (see enlarged curve in Supplementary Fig. 23). All these results underscore the excellent self-compliance of pLCE ionic nanomesh under dynamic deformations, which significantly outperforms common elastic on-skin devices.

## Stress-free muscle-specific electromyography

The combination of self-complaint and permeable properties of pLCE ionic nanomesh highlights its advantages for stress-free on-skin electronics over traditional devices under dynamic conditions. We first evaluated the biocompatibility of pLCE ionic nanomesh. L929 cytotoxicity testing demonstrated >92% cell viability after 72 h across varying extract concentrations (Supplementary Figs. 24 and 25). To collect accurate electrophysiological signals, we then prepared pLCE ionic nanomesh electrodes by screen-printing stretchable silver conductors directly onto the nanomesh. The nanomesh's porous structure facilitated the penetration of the conducting ink, creating an intimate adhesion crucial for conformal deformation (Fig. 6a). This hybrid electrode effectively conducts mixed ion-electron signals, forming a non-faradaic junction that couples the ionic current from the skin with the electronic current from the apparatus[49]. Owing to its high conductivity and compliance, the pLCE ionic nanomesh electrode exhibited significantly lower interfacial impedance than commercial Ag/AgCl gel electrodes across a wide frequency spectrum (10⁻¹–10⁶ Hz) (Fig. 6b). This property reduces resistive path limitations while diminishing capacitive reactance hindrance, thereby greatly facilitating bioelectrical signal transmission. Furthermore, the hybrid electrode retained the inherent self-compliance of pLCE ionic nanomesh, allowing it to adapt perfectly to highly curved surfaces like fingerprint and wrinkled skin (Fig. 6c).

Leveraging these properties, we applied the pLCE ionic nanomesh electrode to detect muscle-specific EMG signals on hands for gesture recognition (Fig. 6d). This is a challenging task for traditional electrodes, which typically exhibit low self-compliance. Indeed, commercial Ag/AgCl gel electrodes cannot adhere effectively to the highly curved hand skin, leading to significant motion artifacts and even interfacial failure during signal collection (Fig. 6e, Supplementary Fig. 26). In contrast, the self-compliant pLCE ionic nanomesh

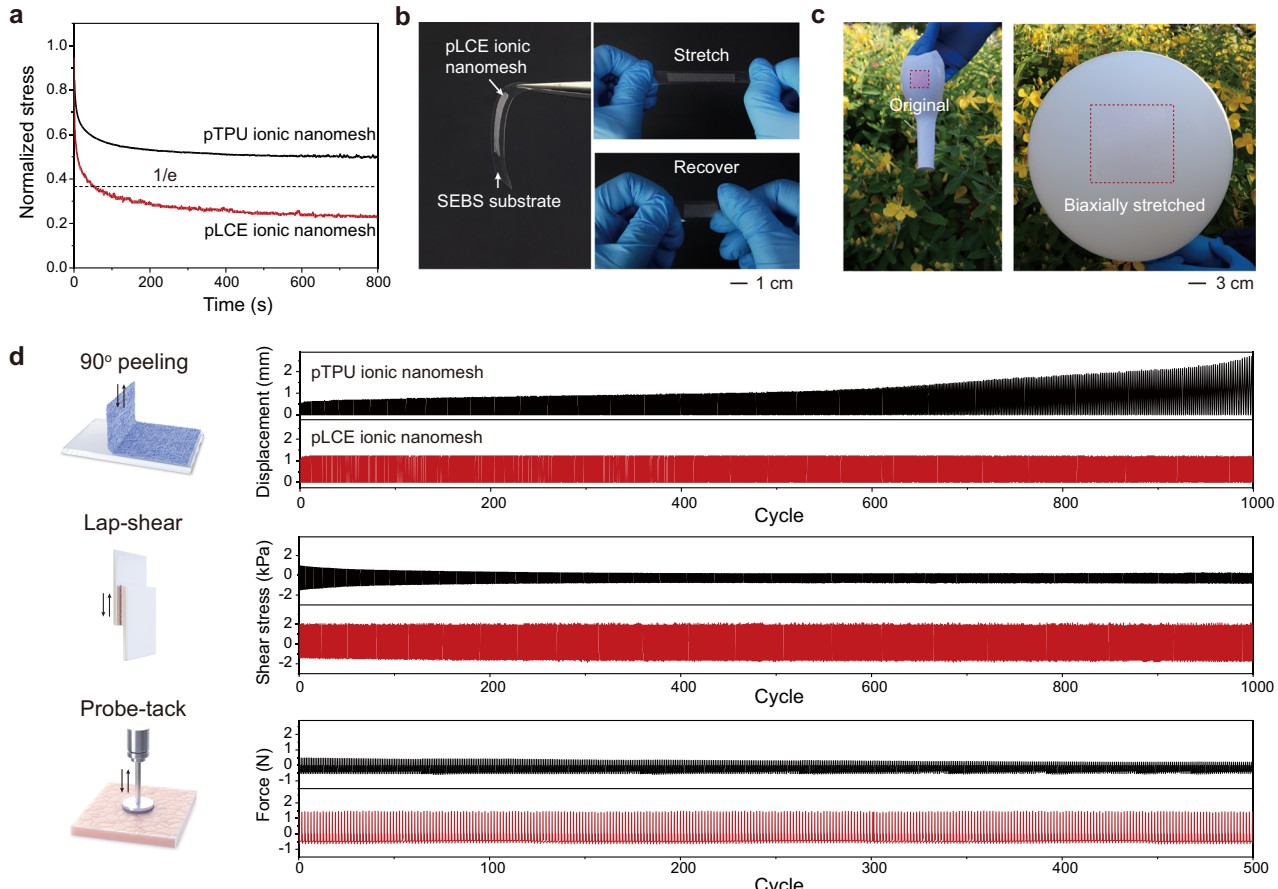

**Fig. 5 | Self-compliance of pLCE ionic nanomesh. a** Stress relaxation curves of permeable thermoplastic polyurethane (pTPU) and pLCE ionic nanomeshes measured at 200% tensile strain. **b** Photographs of pLCE ionic nanomesh attached to a styrene-ethylene-butylene-styrene (SEBS) block copolymer substrate during stretching. **c** Self-compliance of a pLCE ionic nanomesh (sheath dyed by Rhodamine B) on an inflated, biaxially stretched balloon. **d** Cyclic 90° peeling, lap-shear, and probe-tack tests of pTPU and pLCE ionic nanomeshes. 90° peeling was performed on glass substrates, while lap-shear and probe-tack tests were performed on porcine skin substrates.

electrodes form conformal adhesion with the skin of the hand, even amidst dexterous movements (Fig. 6f).

For hand muscle-specific EMG monitoring, we positioned two pLCE ionic nanomesh electrodes over the flexor digitorum superficialis and extensor digitorum muscles, with a reference electrode placed on the dorsum of the hand. EMG signals were then collected to capture evoked action potentials during subtle thumb movements, including moving up and down, left and right, and rotating around (Fig. 6g). These different thumb movements generated unique and reproducible action potential signals, as confirmed by the high-quality time-resolved waveforms, with all signal-to-noise ratios (SNR) exceeding 25 dB (Supplementary Fig. 27).

Finally, we demonstrated the promising applications of the pLCE ionic nanomesh electrode in detecting hand motions in practical scenarios. Real-time EMG signals were collected as a hand grasped various objects, including an apple, large and small soft balls, and even a vibrating massage ball (Fig. 6h; SNRs > 27 dB, Supplementary Fig. 28). Once again, our self-compliant electrodes captured distinct yet reproducible signals. Crucially, the electrodes maintained high signal fidelity even under active vibration, showcasing their significant potential to suppress motion artifacts under dynamic conditions. We further demonstrated that the pLCE ionic nanomesh electrode could also convey high-quality EMG signals on sweaty skin, owing to the water stability of the self-compliant LCE core (Supplementary Fig. 29). Moreover, applying the nanomesh electrodes to collect electrocardiogram (ECG) signals highlighted its promising applications in a broader range of electrophysiological monitoring scenarios (Supplementary Fig. 30).

## Discussion

In summary, we have developed an ultrathin, sheath-core ionic nanomesh featuring an LCE core that exhibits unique liquid-like soft elasticity. This design effectively reconciles self-compliance with gas permeability, enabling motion artifact-free on-skin electronics. The soft elasticity of the LCE is crucial for maximizing stress relaxation at the skin-device interface, allowing the ionic nanomesh to adhere and deform seamlessly with skin movement. Furthermore, an ultrasoft and hygroscopic ionic sheath coating on the LCE core provides excellent ionic conductivity and skin biocompatibility. The resulting ionic nanomesh exhibits superior compliant adhesion on skin, demonstrating remarkable resistance to various interfacial failure modes and outperforming conventional on-skin devices. These combined attributes enable the ionic nanomesh electrodes to accurately monitor subtle, muscle-specific hand motions, delivering high-quality EMG signals even under challenging vibrational conditions. We believe this self-compliant nanomesh design represents a transformative platform for advanced bioelectronics, offering both high signal fidelity and wearable comfort.

## Methods
### Materials
1,4-Bis-[4-(3-acryloyloxypropyloxy) benzoyloxy]-2-methylbenzene (RM257) was purchased from Chem Greatwall. 2,2′-(Ethylenedioxy)

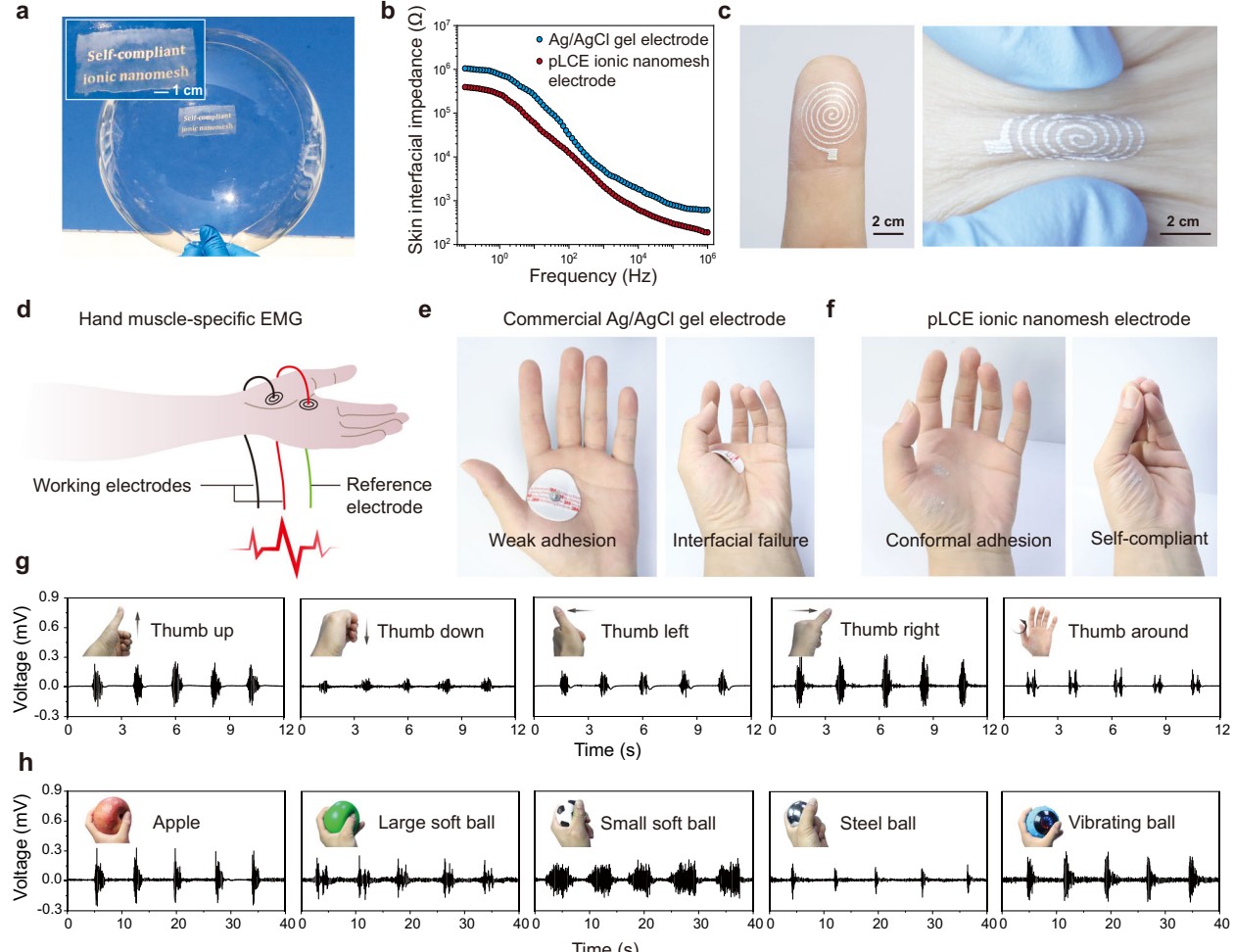

**Fig. 6 | Muscle-specific electromyography (EMG) applications of pLCE ionic nanomesh. a** Photograph of a pLCE ionic nanomesh with printed silver conductor patterns adhered to an inflated balloon, demonstrating their conformal deformation. **b** Skin interfacial impedance of pLCE ionic nanomesh and commercial Ag/AgCl gel electrodes. **c** Photographs of pLCE ionic nanomesh electrodes adhered to a highly curved fingerprint and wrinkled skin. **d** Schematic illustration of hand muscle-specific EMG monitoring. **e, f** Photographs of a commercial Ag/AgCl gel electrode and pLCE ionic nanomesh electrodes on a moving hand. **g** Real-time EMG signals recorded with pLCE ionic nanomesh electrodes during various thumb movements (up, down, left, right, and rotating around). **h** Real-time EMG signals collected by repeatedly grasping an apple, large and small soft balls, a steel ball, and a vibrating ball.

diethanethiol (EDDET), polyvinylpyrrolidone (PVP, $M_w$ = 360,000 g mol[-1]), and thermoplastic polyurethane (TPU, catalog No. 430218) were purchased from Sigma-Aldrich. Dipropylamine (DPA) was purchased from TCI Chemical. Acetylcholine chloride (AChCl) and benzophenone were purchased from Adamas. 1-Butyl-3-methylimidazolium hexafluorophosphate (BMIM PF$_6$), dimethylformamide (DMF), tetrahydrofuran (THF), and ethanol were purchased from Aladdin. Coumarin 6 and methylene blue were purchased from Macklin. Elec-N50 conductive silver paste was purchased from Shenzhen Yi Lai Technology Co., Ltd. Commercial Ag/AgCl gel electrode was obtained from 3 M Company. All chemicals were used as received without further purification.

**Preparation of LCE spinning dope**

RM257 (4 mmol, 2.4 g) was first dissolved in a mixed solvent of DMF and THF (9 mL, 8:1 v/v) at 80 °C to ensure complete dissolution. Subsequently, EDDET (4 mmol, 0.768 g), TPU (0.19 g, pre-dissolved in DMF with a concentration of 10 wt%), and benzophenone (0.009 g) were sequentially added to the solution. To catalyze the Michael addition reaction, 12 μL of diphenylamine (DPA) was introduced. The mixture was then continuously stirred at room temperature (25 °C) for 12 h until a homogeneous spinning dope was obtained. The resulting

spinning dope was stored in a brown glass bottle, protected from light, to maintain its stability.

To prepare LCE film for rheological and tensile tests, the spinning dope was injected into a sandwich glass mold lined with two release films and separated by a 1 mm-thick silicone spacer. The mold was then exposed to UV light (365 nm, 100 W) for 30 min to initiate crosslinking. Finally, the sample was dried in a vacuum oven at 80 °C for 72 h to remove the solvent.

**Preparation of PVP/AChCl spinning dope**

PVP (4 g), AChCl (1.2 g), and benzophenone (0.02 g) were dissolved in a mixure of DMF and ethanol (20 mL, 9:1 v/v). The solution was magnetically stirred for 2 h at room temperature until it became a homogeneous and transparent liquid. The spinning dope was stored in a brown glass bottle, protected from light. For CLSM observation, Coumarin 6 (0.1 wt%) was added to the dope for fluorescent labeling.

To prepare a PVP/AChCl film for permeability and tensile tests, PVP and AChCl were dissolved in ethanol. The solution was injected into a sandwich glass mold lined with two release films and separated by a 0.1 mm-thick silicone spacer. The mold was exposed to UV light (365 nm, 100 W) for 30 min to initiate crosslinking. Finally, the sample

was dried under ambient conditions (25 °C, RH 50%) until moisture equilibrium was reached.

## Preparation of TPU spinning dope

TPU (1 g) was dissolved in a mixed solvent of DMF and THF (9 g, 1:1 w/w) to obtain a homogeneous solution with a polymer concentration of 10 wt%. To prepare TPU films for rheological and tensile tests, the spinning dope was injected into a 1 mm-thick silicone mold, and then dried in a vacuum oven at 80 °C for 72 h.

## Fabrication of pLCE/pTPU ionic nanomeshes

pLCE ionic nanomesh was fabricated by coaxial electrospinning with a coaxial stainless-steel needle (18 G for the outer phase and 25 G for the inner phase). The outer fluid consisted of the PVP/AChCl spinning dope, with a flow rate ranging from 0–1.2 mL h$^{-1}$. The inner fluid was the LCE spinning dope, maintained at a flow rate of 0.3 mL h$^{-1}$. We applied a voltage of ~12 kV, a collection rotation speed of 10 rpm, and a tip-to-collector distance of 20 cm to achieve a stable Taylor cone. A UV spot curing system (OmniCure S1500, 30 W cm$^{-2}$) was used to induce chemical crosslinking during electrospinning. All samples were deposited onto a thin release paper at 25 °C and RH 40%.

For pTPU ionic nanomesh, the fabrication process was identical to that of pLCE ionic nanomesh, except the inner fluid was replaced with the TPU spinning dope (sheath flow rate: 0.4 mL h$^{-1}$).

For LCE ionogel nanomesh, BMIM PF$_6$ (50 mol% relative to the EDDET spacer) was added to the LCE spinning dope for electrospinning without a sheath flow.

## Fabrication of pLCE ionic nanomesh electrode

We employed screen printing to create stretchable Ag electrode patterns directly on the pLCE ionic nanomesh. During printing, the nanomesh was first adhered to a release paper and securely fixed to prevent movement. The Elec-N50 conductive silver paste (containing 50 wt% nanosilver, 15 wt% water-borne polyurethane, and 30 wt% water; viscosity: 10–25 Pa s) as the printing ink was then screen-printed onto the nanomesh with the aid of a mask. The patterned circuits were cured for 12 h at room temperature to ensure electrical conductivity and adhesion.

## Characterizations

The morphologies of nanomeshes were examined by a scanning electron microscope (SEM, Hitachi Regulus 8230) at an acceleration voltage of 10 kV. TEM and elemental mapping images were acquired on a transmission electron microscope (JEM-2100F, JEOL). POM images were taken on a polarized optical microscope (BX53P, Olympus). Confocal laser scanning microscopy (CLSM) images were captured by a Leica SP8 microscope. FTIR spectra were collected in the attenuated total reflectance (ATR) mode on a Nicolet iS50 FTIR spectrometer. Contact angles were measured on a contact angle goniometer (OCA15EC, Dataphysics). Ionic conductivity and electrochemical impedance were measured using an electrochemical workstation (CHI760E, Chenhua). Gel permeation chromatography (GPC) was performed on a Waters ACQUITY APC system using THF as the eluent.

## Tensile test

Mechanical properties were evaluated using a vertical dynamometer (ESM303, MARK-10) equipped with a 1 N load cell. For these tests, the nanomesh was mounted on an auxiliary pre-cut PET substrate. Both tensile curves and stress relaxation curves were recorded at room temperature (25 °C) and RH 50%. The strain rate was set to 0.1 s$^{-1}$.

## Low-field $^1$H NMR measurement

Solid-state low-field NMR measurements were conducted on a VTMR20-010V-I NMR analyzer (Suzhou Niumag Analytical Instrument Corporation). A $^1$H NMR probe was used to measure the $T_2$ profiles and

2D $T_1$-$T_2$ maps of pLCE ionic nanomeshes. A Q-CPMG sequence was selected for 1D low-field NMR measurement and a Q-SR-CPMG sequence was selected for 2D low-field NMR measurement.

## Permeability test

To assess moisture permeability, each membrane was mounted onto a glass bottle containing 5 g of deionized water. After sealing, the samples were tested for 12 h at 25 °C and RH 50%. The bottles were weighed every 2 h, and the water vapor transmission rate (WVTR) was calculated using the formula: WVTR = $(w_0\text{-}w_1)/S*t$, where $w_0$ and $w_1$ refer to the weights before and after a period $t$, and $S$ is the test area of the membrane.

Air permeability was evaluated using a YG461E-III (Wenzhou Fangyuan Instrument Co., Ltd) fully automatic permeability tester, in accordance with the GB/T 5453-1997 standard. Tests were performed on a 20 cm$^2$ sample under controlled conditions (25 °C, RH 50%). Pressure differentials across the sample were applied in 25 Pa increments, ranging from 50 to 200 Pa, with permeability measurements recorded at each interval.

## Rheological test

Films with a thickness of 1 mm were fabricated for rheological tests. Small-amplitude oscillatory shear tests, specifically for generating time-temperature superposition curves, were conducted using an ARES-G2 rheometer equipped with an 8-mm parallel-plate geometry. A constant oscillatory strain amplitude of 0.1% was applied during frequency sweeps conducted from 0.1 to 100 Hz, with an axial force of 0.5 N maintained throughout the testing. Temperature-sweep rheological tests were conducted from -40 to 120 °C at a heating rate of 5 °C min$^{-1}$. A constant oscillatory strain amplitude of 0.1% and a frequency of 0.1 Hz were applied during the temperature sweep.

## 90° peeling test

Interfacial toughness was measured by the 90° peeling method with a vertical dynamometer (ESM303, MARK-10). Prior to testing, the pLCE ionic nanomesh (1 cm in width, 6 cm in length) with a double-sided tape as the stiff back was adhered to different substrates. The peeling speed was set to 50 mm min$^{-1}$.

## Cyclic 90° peeling test

To quantify the fatigue-resistant adhesion performance of the pLCE and pTPU ionic nanomeshes, we conducted 90° peeling tests under cyclic loading (force control mode) using the same vertical dynamometer (ESM303, MARK-10) setup. In these multiple-cycle peeling tests, we applied a cyclic peeling force with amplitudes of 0.2, 0.4, 0.6 and 0.8 N. The corresponding interfacial crack extension was continuously recorded directly from the testing machine.

## Cyclic lap-shear test

Adhesive specimens (25 × 20 mm$^2$) were sandwiched between porcine skin substrates. Experiments were conducted on a vertical dynamometer (ESM303, MARK-10). Displacement-controlled tests were performed at a speed of 100 mm min$^{-1}$ with a strain amplitude of ±5%.

## Cyclic probe-tack test

pLCE or pTPU ionic nanomeshes were attached to the upper plate of vertical dynamometer (ESM303, MARK-10) as the probe. The probe was then lowered onto porcine skin at a rate of 20 mm min$^{-1}$, maintained in contact under a 0.5 N force for 30 s, and then separated at a lifting rate of 20 mm min$^{-1}$.

## Cytotoxicity test

Cytocompatibility was assessed via CCK-8 assay using L929 fibroblasts. Leaching solutions (0.5-1.5 mg mL$^{-1}$) of the pLCE ionic nanomesh were prepared by UV-sterilizing samples and incubating them in DMEM for

72 h. Cells were seeded in 96-well microplates at a density of $1 \times 10^4$ cells per well and then cultured in 5% $CO_2$ at 37 °C for 24 h. The cells were then exposed to the extracts for 24, 48, and 72 h, respectively, followed by CCK-8 treatment (4 h, 37 °C). Absorbance at 450 nm was measured to quantify viability relative to controls.

### Electrophysiological measurement
Muscle-specific EMG signals were recorded from the hand using a commercial device (EMG06, Borunyin Technology Co., Ltd.) in a triple-electrode configuration at a sampling rate of 330 Hz. Two pLCE ionic nanomesh electrodes served as working electrodes and were attached to the flexor digitorum superficialis and extensor digitorum muscles. A third nanomesh electrode was placed on the dorsal side of the hand as a reference. The electrodes were connected to the EMG device using flexible copper foil and wire. EMG signals were then recorded to measure evoked action potentials during subtle thumb movements. ECG signals were recorded using a commercial device (STM32, Borunyin Technology Co., Ltd.) in a triple-electrode configuration at a sampling rate of 250 Hz. Two pLCE ionic nanomesh electrodes served as working electrodes and were attached to the inner sides of the left and right arms. A third nanomesh electrode was placed on the right lower abdomen as a reference. A volunteer participated in the experiments as approved by the Institutional Biomedical Research Ethics Committee of Donghua University (No. SRSY202509060079). The written and informed consent was received from the volunteer.

### Statistics and reproducibility
All experiments were repeated independently for at least three times with similar results.

### Reporting summary
Further information on research design is available in the Nature Portfolio Reporting Summary linked to this article.

## Data availability
All data supporting the findings of this study are available within this article and Supplementary Information or from the corresponding authors upon request. The data generated in this study are provided in the Supplementary Information/Source Data file. Source data are provided with this paper.

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

## Acknowledgements

We gratefully acknowledge the financial supports from the National Natural Science Foundation of China (NSFC) (Nos. 52322306, 22275032 received by S.S.; No. 52433003 received by P.W.). S.S. appreciates the support from Shanghai Oriental Talent Program.

## Author contributions

Q.D. carried out most experiments and co-wrote the manuscript. L.L. performed cytotoxicity test. S.S. and P.W. supervised the project and co-wrote the manuscript. All authors discussed the results and revised the manuscript.

## Competing interests

The authors declare no competing interests.
