## [Transparent Peer Review file · Nature Communications]

Self-compliant ionic nanomesh for gas-permeable and stress-free on-skin electronics

Corresponding Author: Professor Peiyi Wu

Version 0:

Reviewer comments:

Reviewer #1

(Remarks to the Author)

This manuscript presents an interesting design for on-skin bioelectronics by integrating self-adaptive compliance and gas permeability into a single platform, enabling its effective operation under dynamic skin deformations. A coaxial electrospinning technique was used to create an ultrathin ionic nanomesh with a self-compliant LCE core and a soft ionic sheath. The resulting nanomesh electrodes show great promise to capture high-quality muscle-specific EMG signals. The whole manuscript is well organized and supported by comprehensive experimental data. I think this work is suitable for publication in Nature Communications by addressing the following concerns.

1. The authors have compared the moisture permeability of the nanomeshes at different sheath flow rates and highlighted their superior performance over PDMS films. To further contextualize this finding, it would be beneficial to also include a comparison of the water vapor transmission rate of ionic nanomesh with those reported for other similar systems in the literature.
2. For flexible and on-skin materials, fracture resistance, and specifically notch insensitivity, are critical mechanical properties. Please discuss on the notch resistance of the electrospun nanomesh. This would help readers better evaluate the practical durability of the device and its robustness.
3. LCEs are known to possess reversible actuation capabilities. It would be helpful if the authors could show whether this key property is preserved after LCE has been processed into the coaxial electrospun nanomesh structure.
4. The manuscript contains several minor spelling and typographical errors. For instance, there appear to be inaccuracies in the labels or annotations in Supplementary Fig. 13a. A thorough proofreading of all figure captions and supplementary materials is recommended to ensure linguistic accuracy and clarity.

Reviewer #2

(Remarks to the Author)

In this work, Du et al. presented an innovative design of a liquid crystal elastomer-based sheath–core ionic nanomesh, which cleverly exploited the “soft elasticity” of liquid crystal elastomers to achieve significant stress relaxation. By combining a hydrophilic sheath layer (PVP/ACHCl) with a porous architecture, the authors also realized high permeability, thereby effectively addressing the fundamental trade-off between self-adaptability and permeability that has long constrained conventional on-skin electronic devices. In my view, this work holds considerable value both in terms of its fundamental scientific insights and its meticulous engineering execution. I therefore strongly recommend its publication in Nature Communications. Some minor revisions may further strengthen the presentation and impact of their scientific findings.

Comments:

1. UV-assisted coaxial electrospinning process was used in the preparation process. The authors systematically varied the sheath flow rate to determine the optimal processing conditions, ultimately identifying 0.4 mL h^{-1} as the optimal sheath flow rate. Could they further clarify what kind of material can be obtained if the sheath flow rate exceeds 1.2 mL h^{-1} ?
2. From the temperature-sweep rheology curves, a significant reduction in the magnitude of the second $\tan \delta$ peak was observed upon the direct incorporation of the ionic liquid BMIMPF₆. However, the authors primarily present this result without providing an explanation for the underlying cause of this notable decrease.
3. How about the scratch resistance of the ionic nanomesh when adhered on skin? This is important for long-term skin-mounted applications.

4. The authors mainly selected muscle-specific electromyography (EMG) for monitoring precise hand movements. I suggest adding electrocardiogram (ECG) tests of pLCE ionic nanomesh electrodes in both the resting and motion states for further demonstrating the self-compliant advantage of the ionic nanomesh.
5. It is recommended that the authors provide the full term for each abbreviation upon its first use in the text. This would significantly improve readability.

Reviewer #3

(Remarks to the Author)

In this paper, the authors presented a coaxial electrospinning strategy to fabricate an ultrathin LCE-based ionic nanomesh that effectively combines self-compliance and gas permeability. The nanofibers are featured with a sheath-core structure for capturing high-quality, muscle-specific EMG signals and allowing for the precise classification of even very subtle thumb actions. The discussion is relatively comprehensive, I think additional major revision is necessary for publication as a full paper.

Here are my suggestions:

- 1、 How about the orientation property of the electrospinning nanofibers? The authors are also suggested to give more discussions on the influence derived from the high-voltage.
- 2、 The authors optimized the sheath flow rate during the fabrication process. I wondered more about the influence of core flow rate. The discussions on the ratio of core flow rate and sheath flow rate change are also expected.
- 3、 As shown in Fig. 3d, the pLCE ionic nanomesh was demonstrated with good water permeability. How about the permeability of the oily liquid?
- 4、 How about the self-compliant property of pLCE ionic nanomesh electrode on the wetted surface? If the skin is in a sweating state, will the large amount of sweat on the surface affect the sensing of electrical signals?
- 5、 The English expression should be completely revised by a native speaker.

Version 1:

Reviewer comments:

Reviewer #1

(Remarks to the Author)

It has been well revised, I suggest accept it

Reviewer #2

(Remarks to the Author)

The authors have thoroughly revised the manuscript in response to the reviewer's comments, leading to a significant improvement in the overall quality of the work. The reviewer is satisfied with the revisions and has no further suggestions. I recommend that the manuscript be accepted for publication.

Reviewer #3

(Remarks to the Author)

The paper has been well improved. All of my concerns are addressed with additional discussions concerning several obscures provided. Some details are also included in the revised manuscript. I am therefore pleased to see it published as its current form.

Response to Reviewers

Reviewer #1:

This manuscript presents an interesting design for on-skin bioelectronics by integrating self-adaptive compliance and gas permeability into a single platform, enabling its effective operation under dynamic skin deformations. A coaxial electrospinning technique was used to create an ultrathin ionic nanomesh with a self-compliant LCE core and a soft ionic sheath. The resulting nanomesh electrodes show great promise to capture high-quality muscle-specific EMG signals. The whole manuscript is well organized and supported by comprehensive experimental data. I think this work is suitable for publication in Nature Communications by addressing the following concerns.

Response: We sincerely thank the Reviewer for the time and effort in evaluating our manuscript (NCOMMS-25-64646-T) submitted to *Nature Communications*, as well as for the highly constructive comments. In response to the Reviewer's suggestions, we have carefully revised our manuscript. Below, we provide a point-by-point response to all queries. We hope our revisions adequately address the concerns raised and would be grateful for a re-evaluation of the revised manuscript.

Comments:

1. The authors have compared the moisture permeability of the nanomeshes at different sheath flow rates and highlighted their superior performance over PDMS films. To further contextualize this finding, it would be beneficial to also include a comparison of the water vapor transmission rate of ionic nanomesh with those reported for other similar systems in the literature.

Response: We thank the Reviewer for this valuable suggestion. As recommended, we have added a Supplementary Table S1 comparing the water vapor transmission rate (WVTR) of our ionic nanomesh with other typical breathable on-skin materials. Notably, our pLCE ionic nanomesh achieved a WVTR of $1245 \text{ g m}^{-2} \text{ day}^{-1}$ with a thickness of only $8 \text{ }\mu\text{m}$, which is comparable to the best-performing breathable on-skin materials reported in the literature.

Supplementary Table 1. Comparison of moisture permeability among typical breathable on-skin materials.

Materials	Thickness (μm)	WVTR ($\text{g m}^{-2} \text{day}^{-1}$)	Ref.
pLCE ionic nanomesh	8	1245	This work
MXene/PDMS/PDA/PU nanocomposite	~200	1026	[S2]
PAAm-alginate hydrogel	50	1890	[S3]
EGaIn-SBS mat	320	724	[S4]
PVA hydrogel nanomesh	6	~2000	[S5]
PU-gelatin hydrogel	2.7	1252.3	[S6]
liquid-metal-containing nanofibre membrane	21	2941	[S7]
3D liquid diode	650	~1700	[S8]
Wet-adaptive electronic skin	~300	852	[S9]
IP6-PVA ion-conductive elastomer	150	448.8	[S10]

Corresponding revisions can be found in the highlighted text on Pages 8 and S18.

2. For flexible and on-skin materials, fracture resistance, and specifically notch insensitivity, are critical mechanical properties. Please discuss on the notch resistance of the electrospun nanomesh. This would help readers better evaluate the practical durability of the device and its robustness.

Response: We thank the Reviewer for this suggestion. We have supplemented notch resistance measurements by stretching notched and unnotched pLCE ionic nanomeshes, as shown in Supplementary Fig. 19. The ionic nanomesh with a 1/5 notch could still be stretched to over 200% strain, indicating its high notch resistance and the ability to tolerate mechanical damage. This is primarily due to the nanofibrous network structure, which dissipates energy effectively through restricted fibril shearing and controlled slippage, similar to natural silk materials (Greiner, et al. *Science* **2019**, 366, 1376; Ling,

Supplementary Fig. 19. Notch resistance of pLCE ionic nanomesh.

Corresponding revisions can be found in the highlighted text on Pages 11 and S12.

3. LCEs are known to possess reversible actuation capabilities. It would be helpful if the authors could show whether this key property is preserved after LCE has been processed into the coaxial electrospun nanomesh structure.

Response: We thank the Reviewer for this valuable comment. Indeed, heat-induced actuation is a well-known property of LCEs. To confirm its preservation, we conducted an additional experiment by heating a stretched pLCE ionic nanomesh, as shown in Supplementary Fig. 12. The ionic nanomesh rapidly contracted by ~17% when heated to 80 °C, confirming that the actuation property is retained. This behavior results from the heat-induced liquid crystal-to-isotropic transition of the LCE, generating a contractive force to restore its original length. This observation also confirms the ambient liquid crystalline state of the LCE core in the ionic nanomesh.

Supplementary Fig. 12. Heat-induced contraction of stretched pLCE ionic nanomesh.

Corresponding revisions can be found in the highlighted text on Pages 6 and S8.

4. The manuscript contains several minor spelling and typographical errors. For instance, there appear to be inaccuracies in the labels or annotations in Supplementary Fig. 13a. A thorough proofreading of all figure captions and supplementary materials is recommended to ensure linguistic accuracy and clarity.

Response: We thank the Reviewer for the careful review and valuable feedback. We have thoroughly proofread the manuscript, including all figure captions and supplementary materials, and corrected spelling and typographical errors (highlighted with a green background). Specifically, the inaccuracies in Supplementary Fig. 13a (now Supplementary Fig. 20) have been addressed and corrected. We appreciate your attention to detail and have ensured that all textual elements now meet the required standards of linguistic accuracy and clarity.

Reviewer #2:

In this work, Du et al. presented an innovative design of a liquid crystal elastomer-based sheath–core ionic nanomesh, which cleverly exploited the “soft elasticity” of liquid crystal elastomers to achieve significant stress relaxation. By combining a hydrophilic sheath layer (PVP/AChCl) with a porous architecture, the authors also realized high permeability, thereby effectively addressing the fundamental trade-off between self-adaptability and permeability that has long constrained conventional on-skin electronic devices. In my view, this work holds considerable value both in terms of its fundamental scientific insights and its meticulous engineering execution. I therefore strongly recommend its publication in Nature Communications. Some minor revisions may further strengthen the presentation and impact of their scientific findings.

Response: We sincerely thank the respected Reviewer for the time and effort in evaluating our manuscript (NCOMMS-25-64646-T) submitted to *Nature Communications*, as well as for the positive and constructive feedback. In accordance with the Reviewer’s suggestions, we have carefully revised the manuscript. Below is our point-by-point response to all comments. We hope we have adequately addressed the Reviewer’s concerns and would be grateful for a re-evaluation of the revised manuscript.

Comments:

1. UV-assisted coaxial electrospinning process was used in the preparation process. The authors systematically varied the sheath flow rate to determine the optimal processing conditions, ultimately identifying 0.4 mL h⁻¹ as the optimal sheath flow rate. Could they further clarify what kind of material can be obtained if the sheath flow rate exceeds 1.2 mL h⁻¹?

Response: We thank the Reviewer for this valuable suggestion. We optimized the ionic nanomesh mainly based on three criteria: adhesion, gas permeability, and electrospinning quality. We found that further increasing the sheath flow rate to 1.4 mL h⁻¹ resulted in an excessively thick sheath layer, leading to a nearly complete loss of nanomesh porosity, as shown in the added Supplementary Fig. 7. Therefore, we did

not include sheath flow rates higher than 1.2 mL h^{-1} in our detailed discussion.

Supplementary Fig. 7. SEM image of the ionic nanomesh produced at a sheath flow rate of 1.4 mL h^{-1}

Corresponding revisions can be found in the highlighted text on Pages 6 and S6.

2. From the temperature-sweep rheology curves, a significant reduction in the magnitude of the second $\tan \delta$ peak was observed upon the direct incorporation of the ionic liquid BMIMPF₆. However, the authors primarily present this result without providing an explanation for the underlying cause of this notable decrease.

Response: We sincerely thank the Reviewer for this valuable comment. We indeed observed a significant reduction in the second $\tan \delta$ peak, characteristic of soft elasticity, upon introducing BMIM PF₆ (Supplementary Fig. 6). According to our previous study (Ref. S1: Wu, et al. *Adv. Mater.* **2021**, 33, 2103755), this is mainly attributed to the plasticizing effect of BMIM PF₆, which selectively binds to the soft spacers of the LCE and partially disrupts the collective assembly of mesogens. Consequently, the soft elasticity arising from mesogen reorientation is diminished.

Corresponding revisions can be found in the highlighted text on Page S5.

3. How about the scratch resistance of the ionic nanomesh when adhered on skin? This is important for long-term skin-mounted applications.

Response: We thank the Reviewer for this suggestion. As recommended, we evaluated the scratch resistance of the adhered pLCE ionic nanomesh on skin by repeated rubbing with a finger. As shown in the added Supplementary Fig. 21, the ionic nanomesh remained intact without delamination, demonstrating its good scratch resistance.

Supplementary Fig. 21. Scratch resistance of pLCE ionic nanomesh adhered on hand skin.

Corresponding revisions can be found in the highlighted text on Pages 13 and S13.

4. The authors mainly selected muscle-specific electromyography (EMG) for monitoring precise hand movements. I suggest adding electrocardiogram (ECG) tests of pLCE ionic nanomesh electrodes in both the resting and motion states for further demonstrating the self-compliant advantage of the ionic nanomesh.

Response: We thank the Reviewer for this suggestion. We have supplemented electrocardiogram (ECG) results using pLCE ionic nanomesh electrodes in Supplementary Fig. 30. Highly reliable data were obtained under resting, walking, and running conditions, demonstrating the reliability of our electrodes for collecting ECG signals. This highlights the potential applications of our self-compliant ionic nanomesh electrodes in a broader range of electrophysiological monitoring scenarios.

Supplementary Fig. 30. Real-time ECG monitoring recorded with pLCE ionic nanomesh electrodes under resting, walking, and running conditions.

Corresponding revisions can be found in the highlighted text on Pages 16, 22, and S17.

5. It is recommended that the authors provide the full term for each abbreviation upon its first use in the text. This would significantly improve readability.

Response: We greatly thank the Reviewer for this suggestion. We have checked the entire manuscript and provided the full terms of all abbreviations upon their first appearance. Corresponding revisions are highlighted with a yellow background.

Reviewer #3:

In this paper, the authors presented a coaxial electrospinning strategy to fabricate an ultrathin LCE-based ionic nanomesh that effectively combines self-compliance and gas permeability. The nanofibers are featured with a sheath-core structure for capturing high-quality, muscle-specific EMG signals and allowing for the precise classification of even very subtle thumb actions. The discussion is relatively comprehensive, I think additional major revision is necessary for publication as a full paper.

Response: We sincerely thank the respected Reviewer for the time and effort in evaluating our manuscript (NCOMMS-25-64646-T) submitted to *Nature Communications*, as well as for the positive and constructive feedback. In accordance with the Reviewer's suggestions, we have carefully revised the manuscript. Below is our point-by-point response to all comments. We hope we have adequately addressed the Reviewer's concerns and would be grateful for a re-evaluation of the revised manuscript.

Here are my suggestions:

1、 How about the orientation property of the electrospinning nanofibers? The authors are also suggested to give more discussions on the influence derived from the high-voltage.

Response: We thank the Reviewer for this valuable comment.

Typically, fiber orientation is achieved using a high collector rotation speed. In our study, we used a low rotation speed of 10 rpm to collect the electrospun nanofibers. No apparent fiber orientation was observed in the resulting ionic nanomesh, as shown in Fig. 2a and f by SEM and POM images. We intentionally avoided introducing fiber orientation to maintain the initial softness of the pLCE ionic nanomesh and prevent anisotropic stress distribution during deformation. To further investigate the effect of collector rotation speed, we performed additional experiments at speeds ranging from 10 to 1000 rpm. As shown in the added Supplementary Fig. 10, although higher speeds induced nanofiber orientation, the rapid deposition led to incomplete UV crosslinking (without sufficient time) and a gradual loss of mesh porosity.

Supplementary Fig. 10. SEM images of pLCE nanomeshes produced at different collector rotation speeds. A rotation speed of 10 rpm was used in this study.

Using a high voltage incurred similar challenges. As shown in the added Supplementary Fig. 9, increasing the voltage to 14 and 16 kV resulted in also excessively fast nanofiber deposition. This led to incomplete UV crosslinking of the LCE core, and the deposited fibers tended to self-fuse, compromising the nanomesh porosity.

Supplementary Fig. 9. SEM images of pLCE nanomeshes produced at different voltages. A voltage of 12 kV was used in this study.

Corresponding revisions can be found in the highlighted text on Page 6 and S7.

2、 The authors optimized the sheath flow rate during the fabrication process. I wondered more about the influence of core flow rate. The discussions on the ratio of core flow rate and sheath flow rate change are also expected.

Response: We thank the Reviewer for this helpful suggestion. To investigate the influence of the core flow rate, we maintained the sheath flow rate at 0.4 mL h^{-1} and varied the core flow rate from 0 to 0.9 mL h^{-1} . As shown in the added Supplementary Fig. 8, increasing the core flow rate led to a thicker LCE core, gradually reducing the areal porosity from 49% to 23%. This trend is similar to the effect of increasing the

sheath flow rate (Fig. 2a, b). In other words, increasing either the sheath or core flow rate increases the respective layer thickness, both resulting in reduced nanomesh porosity. The core-to-sheath flow rate ratio primarily affects the core-to-sheath thickness ratio but does not directly determine the nanomesh porosity.

Supplementary Fig. 8. SEM images and calculated porosities of pLCE nanomeshes produced at different core flow rates.

Corresponding revisions can be found in the highlighted text on Page 6 and Page S6.

3、 As shown in Fig. 3d, the pLCE ionic nanomesh was demonstrated with good water permeability. How about the permeability of the oily liquid?

Response: We thank the Reviewer for this comment. We performed similar experiments to those in Fig. 3d using oily liquids such as ethyl acetate and corn oil. As shown in the added Supplementary Fig. 14, both hydrophobic liquids did not readily permeate the ionic nanomesh, further confirming its hydrophilic surface. This property is important for selectively transporting moisture when the ionic nanomesh is applied to human skin.

Supplementary Fig. 14. Oil permeability of pLCE ionic nanomesh.

Corresponding revisions can be found in the highlighted text on Pages 9 and S9.

4. How about the self-compliant property of pLCE ionic nanomesh electrode on the wetted surface? If the skin is in a sweating state, will the large amount of sweat on the surface affect the sensing of electrical signals?

Response: We appreciate the Reviewer for this insightful comment. To address this, we additionally tested the performance of our pLCE ionic nanomesh electrodes on sweaty skin. As shown in the added Supplementary Fig. 29, the electrodes adhered firmly to sweaty skin and consistently provided high-quality EMG signals. However, we note that under excessive sweating, the nanomesh may eventually lose adhesiveness due to fluid accumulation, which could compromise signal stability. Therefore, the current ionic nanomesh electrodes are more suitable for skin with minimal to moderate perspiration.

Supplementary Fig. 29. Real-time EMG monitoring recorded with pLCE ionic nanomesh electrodes on sweaty skin.

Corresponding revisions can be found in the highlighted text on Page 16 and Page S17.

5. The English expression should be completely revised by a native speaker.

Response: We are grateful to the Reviewer for this suggestion. The English has been professionally polished by checking and revising the grammar, word choice, and sentence structure throughout the manuscript. Corresponding revisions are highlighted with a **green** background.